# Resilience in the Ranks: Trait Mindfulness and Self-Compassion Buffer the Deleterious Effects of Envy on Mental Health Symptoms among Public Safety Personnel

**DOI:** 10.3390/ijerph19105926

**Published:** 2022-05-13

**Authors:** Shadi Beshai, Sandeep Mishra, Justin R. Feeney, Tansi Summerfield, Chet C. Hembroff, Gregory P. Krätzig

**Affiliations:** 1Department of Psychology, University of Regina, 3737 Wascana Parkway, Regina, SK S4S0A2, Canada; tansi.summerfield@rcmp-grc.gc.ca (T.S.); chet.hembroff@rcmp-grc.gc.ca (C.C.H.); gregory.kratzig@rcmp-grc.gc.ca (G.P.K.); 2Department of Management, Lang School of Business and Economics, University of Guelph, Guelph, ON N1G2W1, Canada; sandeep.mishra@uoguelph.ca; 3Department of Management and Marketing, School of Business, Rhode Island College, 600 Mount Pleasant Avenue, Providence, RI 02908, USA; jfeeney@ric.edu

**Keywords:** envy, dispositional mindfulness, self-compassion, resilience, PSP, depression, anxiety, positive and negative emotions

## Abstract

Public safety personnel (PSP) face frequent stressors that increase their risk of developing symptoms of depression and anxiety. In addition to being exposed to potentially traumatic events, PSP trainees may face a compounded risk of developing mental health symptoms, as their training environments are conducive to social comparisons and the resultant painful emotion of envy. Envy is associated with numerous negative health and occupational outcomes. Fortunately, there are several individual difference factors associated with increased emotional regulation, and such factors may offer resilience against the damaging mental health effects of envy. In this study, we examined the interplay between dispositional mindfulness, self-compassion, and dispositional envy in predicting job satisfaction, stress, experience of positive and negative emotions, subjective resilience, and symptoms of depression and anxiety in a sample of police trainees (*n* = 104). A substantial minority of trainees reported clinically significant symptoms of depression (*n* = 19:18.3%) and anxiety (*n* = 24:23.1%) in accordance with the cut-off scores on screening measures. Consistent with hypotheses, dispositional envy was associated with lower job satisfaction, greater stress, and greater anxiety and depression. Furthermore, envy was associated with higher negative emotions, lower positive emotions, and lower subjective resilience. Dispositional mindfulness and self-compassion were associated with greater job satisfaction, lower stress, and reduced symptoms of depression and anxiety. Moreover, mindfulness and self-compassion were both associated with lower negative emotions, higher positive emotions, and subjective resilience. The associations between envy and the relevant job and mental health outcomes were significantly diminished after controlling for mindfulness and self-compassion. This suggests that these protective traits may serve as transdiagnostic buffers to the effects of envy on mental health. The results of this study confirmed the damaging effects of envy and suggested the potential remediation of these effects through the cultivation of mindfulness and self-compassion.

## 1. Introduction

Public safety personnel (PSP, e.g., police officers, correctional workers, dispatchers, firefighters, and paramedics) experience highly stressful events as part of their profession. This increased and ongoing exposure to stressful and potentially traumatic events puts PSP at a higher risk of developing the symptoms of several mental health conditions, compared to members of the general public [1]. While frequent exposure to potentially traumatic events alone can be a risk factor for more severe mental health symptoms, other factors in the work and training environments of PSP can further compound this already high risk. For example, police trainees, in particular, have an especially demanding training regime [2]. Recruits experience long training days (10–11 h), five days a week, over 26 consecutive weeks, with variable training sessions over their weekends. Recruits live in dormitories of 32 people and must reside at the training academy. Conditions include regular hierarchical evaluations; instructors will often make explicit and directed comments about one trainee’s performance, relative to others within their troop. Such conditions are fertile grounds for *envy*—negative feelings of longing, striving, or desire tied to unfavorable social comparisons [3]. Environments typified by high levels of envy, competition, and negative affectivity can exacerbate the already heightened risk of developing symptoms of anxiety, depression, and stress-based conditions among police trainees.

Envy is a stressor in the workplace that is linked with negative outcomes, such as turnover intent, job dissatisfaction, and reduced organizational commitment, among others (reviewed in [4]). Outside of the workplace, dispositional malicious envy has been strongly associated with poorer mental health [3]. In a recent longitudinal study, researchers found that an increase in the experience of envy on a societal level, corresponded, in a lockstep fashion, with an increase in symptoms of psychopathology [5]. In another study, conducted with Chinese adults, researchers found that increased social comparisons were associated with higher symptoms of depression, and that this relationship was fully mediated by the experience of envy [6]. Furthermore, negative affectivity, which is a critical component of envy [7], is associated with an elevated threat response to stressors, which could exacerbate the risk of significant post-traumatic stress symptoms [8]. Unfortunately, despite these suggestive studies, the role of envy in mental health has been relatively neglected. Furthermore, while envy is conceptualized as a negative social emotion [9], very few studies have examined the direct relationship between envy and the experience of negative (e.g., shame, fear, and hostility) and positive (e.g., attentiveness, determination, and inspiration) emotions. Recently, clinical researchers have begun highlighting the need to understand the degree and nature of the relationship between envy and mental health, as well as the urgent need to develop interventions specifically designed to mitigate envy and its effects [10].

The ability to manage the negative social emotions that result from stressful environments, such as envy, varies individually. The capacity for resilience is “the ability to bounce back or recover from stress” ([11], p. 194). Resilience has often been conceptualized as a stable individual difference, including in the organizational literature (e.g., Ref. [12]). Much of this research has focused on protective attitudes, tendencies, and behaviors. However, resilience (adaptation) can only be demonstrated in the face of adversity. Chmitorz et al.’s [13] resilience model, which was developed within a clinical psychology framework, suggested that maladaptive attitudes, behaviors, emotions, and other outcomes (e.g., poorer mental health and job dissatisfaction) are products of the balance of *risk factors* (capacities that decrease the likelihood of adaptation), over *protective factors* (capacities that enhance adaptation; Ref. [14]). When risk factors overwhelm protective factors, people are more vulnerable to negative outcomes. Conversely, when protective factors overwhelm risk factors, people have a greater capacity for resilience, making “bouncing back” from adversity more likely. In this study, we examined the capacity of envy as a risk factor, and the capacities of self-compassion and mindfulness as protective factors, consistent with Chmitorz et al.’s [13] framework.

Two dispositional capacities in particular may serve as transdiagnostic protective factors in the face of heightened adversity: *mindfulness* and *self-compassion*. Trait or dispositional mindfulness describes the ability to attend to present-moment experiences with non-judgmental acceptance [15,16]. Self-compassion describes the ability to be moved by one’s own suffering and the desire to alleviate this suffering [17,18]. Neff argued that self-compassion was comprised of the following three, related subcomponents: self-kindness (as opposed to self-criticism); common humanity (feeling united with others and nature in suffering, as opposed to feeling alienated by painful experiences); and mindfulness (as opposed to avoidance or overidentification with suffering). Dispositional mindfulness is consistently associated with lower depression and anxiety symptoms, lower maladaptive coping (e.g., rumination and catastrophizing), and higher emotional regulation (e.g., emotional stability and reduced reactivity, [19]). Similarly, self-compassion has been robustly associated with reduced symptoms of several psychopathologies [20]; higher metrics of well-being [21]; and even with better physical health and health behaviors [22]. Mindfulness, specifically, has been associated with positive occupational outcomes, including job satisfaction, commitment, and performance [23]. Self-compassion has been studied less frequently than mindfulness in the workplace; however, the available evidence suggests that self-compassion is associated with improved satisfaction and functioning in diverse workplace settings [24,25,26]. Self-compassion has been studied even less frequently among police and other PSP, a gap that we have shed light on in this study.

The literature reviewed above suggests the following: (a) PSP, including PSP trainees, experience job-related stressors that act as risk factors for heightened symptoms of mental health and for lower job satisfaction; (b) in addition to job-specific stressors, training environments are conducive to increased social comparisons and to the experience of envy among recruits, which, in turn, increases the likelihood of experiencing mental health symptoms; and (c) transactional models of resilience suggest that well-being and vocational satisfaction are determined by a balance of risk (e.g., heightened stressors and envy) and protective factors (e.g., dispositional mindfulness and self-compassion). Unfortunately, there is a paucity of research in examining transactional models of resilience in PSP. Furthermore, as highlighted above, a few previous studies have demonstrated the association of envy with poorer mental health; however, these studies are limited in number and scope, and are almost completely absent among PSP. Additionally, there is very little research that establishes a direct relationship between the dispositional capacity to experience envy and the experience of other negative and positive emotional states. Finally, very few studies have examined the direct relationships of envy, mindfulness and self-compassion, with subjective resilience in PSP, specifically, or with any population, generally. Resilience, even examined as a subjective, static construct, is meaningfully associated with higher quality-of-life and other adaptive health outcomes [27]. As conceptualized, envy attenuates resilience, while mindfulness and self-compassion augment it. Accordingly, examining the nature of the cross-sectional relationships of envy, mindfulness, and self-compassion with resilience, offers a simple and very direct test of such hypotheses.

Both mindfulness and self-compassion may serve as transdiagnostic protective factors in the face of heightened stress, and especially in response to such powerful negative environmental influences that facilitate dispositional envy. We examined the specific capacity for *emotional resilience* among Canadian law enforcement recruits, who experience substantial adversity. In the specific context of this study, we defined emotional resilience as the product of a risk factor (susceptibility to envy) and two self-regulatory protective factors (mindfulness and self-compassion).

Following research that indicated the damaging effects of envy on a range of psychological outcomes, we hypothesized the following:

**Hypothesis** **1a** **(H1a).***Envy will be negatively associated with job satisfaction, and positively associated with stress, depression, and anxiety*.

**Hypothesis** **1b** **(H1b).***Envy will be positively associated with the experience of negative emotions and negatively associated with the experience of positive emotions*.

**Hypothesis** **1c** **(H1c).***Envy will be negatively associated with subjective resilience*.

The adaptive dispositional capacities of mindfulness and self-compassion serve as buffers against the negative consequences of envy. Accordingly, we anticipated that these capacities would extend into the work context, and hypothesized the following:

**Hypothesis** **2a** **(H2a).**
*Mindfulness will be positively associated with job satisfaction and negatively associated with stress, depression, and anxiety.*


**Hypothesis** **2b** **(H2b).***Self-compassion will be positively associated with job satisfaction and negatively associated with stress, depression, and anxiety*.

**Hypothesis** **2c** **(H2c).***Both dispositional mindfulness and self-compassion will be negatively associated with negative emotions, and positively associated with positive emotions, and subjective resilience*.

Chmitorz et al.’s [13] transactional model suggested that a surplus of protective factors, over risk factors should confer resilience. Accordingly, we hypothesized the following:

**Hypothesis** **3** **(H3).***Dispositional mindfulness and self-compassion will account for a unique variance in job satisfaction and mental health symptoms, beyond the risk factor of envy*.

A second implication of Chmitorz et al.’s [13] model was that protective factors should “buffer” the associations of risk factors with negative outcomes. Accordingly, we hypothesized the following:

**Hypothesis** **4** **(H4).***The associations between envy and the outcomes of job satisfaction, stress, depression, and anxiety will be diminished after controlling for mindfulness and self-compassion*.

## 2. Materials and Methods

### 2.1. Participants and Procedure

In the present study, we employed a convenience sampling procedure and cross-sectional design. Survey links were provided to 173 trainees in the Royal Canadian Mounted Police (RCMP), which is the federal police service of Canada. Trainees completed measures online in randomized order, as part of a larger mindfulness trial. Participants were in their first or second week of a multi-week training regimen when they completed the measures. We administered questionnaires to study participants who were in groups of 25–32, in internet-enabled computer rooms on the RCMP training campus. The University of Regina’s Research Ethics Board approved the study prior to commencement of study activities. Of the 173 participants who received the survey, 104 (25 identified as women and 79 identified as men: *M*_age_
*=* 29.27, *SD*_age_ = 6.92) provided their informed consent and responses to the majority of scale items and were retained for analyses. Participants were excluded if they failed to respond to 20% or more of the study items. A total of 46 participants (44.2%) identified as European, three as Korean, two as Persian, and two as Punjabi, with various other identifications (e.g., Russian, Japanese, and Latino). The majority (*n* = 57:55%) of participants reported having tried meditation at least once. Of those who tried meditation, 70% reported their meditation skill level as novice, 26.5% as intermediate, and 3.5% as advanced. After completing questionnaires, participants were offered the opportunity to watch a psychoeducational video on depression, anxiety, and stress. Participants were also offered links to several online guided mindfulness meditation and self-compassion exercises and techniques. Finally, at the end of the trial all participants were offered a list of evidence-based, self-help resources, as well as contact information for local mental health services. Data were collected from April–August 2018.

### 2.2. Measures

#### 2.2.1. Job-Related, Mental Health, and Emotional Outcomes

Abridged Job Descriptive Index (JDI): The JDI is an 8-item, self-report measure assessing general job satisfaction. The JDI has shown excellent reliability and validity in a variety of workplace settings [28].

The Patient Health Questionnaire (PHQ-8): The PHQ-8 is an 8-item, widely used instrument for screening and assessing the severity of depression, based on the Diagnostic Statistical Manual IV’s (DSM-IV) criteria for depression [29].

The Generalized Anxiety Disorder Scale (GAD-7): The 7-item GAD-7 is a widely used instrument for screening and assessing the severity of generalized anxiety symptoms [30]. Both PHQ-8 and GAD-7 have been used and validated extensively.

To quantify clinically significant depression and anxiety, we used a cut-off score of 10 or higher on the PHQ-8 and GAD-7 [31,32]. A cut-off score of 10 or higher on these measures has demonstrated good sensitivity and specificity in both depression and anxiety, respectively, and can reliably discriminate between those meeting and not meeting the diagnostic thresholds for such conditions.

The Positive and Negative Affect Schedule Short Form (PNAS): The PNAS is a 10-item, self-report measure of the presence and intensity of five positive (activity, inspiration, determination, alertness, and attentiveness: PNAS-P) emotions and five negative (upset, hostility, nervousness, fear, and shame: PNAS-N) emotions over the last week. The PNAS has demonstrated adequate psychometric properties when used with general population samples [33].

The Brief Resilience Scale (BRS): The BRS is a 6-item, self-report measure designed to assess perceptions of one’s own capacity to “bounce back” after facing adversity. Respondents rated their agreement with each statement (e.g., “I tend to bounce back quickly after hard times”) on a 7-point Likert scale. The BRS has been demonstrated to correlate significantly and meaningfully with coping, mental, and physical health [11].

#### 2.2.2. Risk Factors

The Dispositional Envy Scale (DES): The 8-item DES assesses self-reported differences in the propensity to feel envy as a result of comparisons to others [34].

The Perceived Stress Scale (PSS): The 10-item PSS assesses self-reported stress, experienced over the past month [35]. The scale has been used extensively and has demonstrated excellent reliability and validity [36].

#### 2.2.3. Protective Factors

The Five Factor Mindfulness Questionnaire (FFMQ): The 15-item FFMQ is a brief measure of stable individual differences in mindfulness tendencies. This scale has been shown to be psychometrically rigorous in comparison with the full, 39-item FFMQ and has been recommended for use in circumstances requiring shorter measures [37].

The Self-Compassion Scale (SCS): The 12-item SCS is a brief measure of attitudes toward oneself in the face of difficulty or failure; the 12-item SCS demonstrates near-perfect correlation with the full, 26-item inventory [18].

### 2.3. Data Analysis Plan

The proportion of missing values within each scale for each participant was calculated. If this proportion exceeded 20%, a total score was not computed and it was treated as a missing value; if the missing proportion was less than 20%, mean imputation was used [38,39]. Simulation studies demonstrate error estimates significantly increase after more than 20% of values are missing within-scale [40]. Scale scores were tabulated so that a higher score on each of the measures was indicative of greater levels of the construct assessed.

To address hypotheses H1 (a–c) and H2 (a–c), we conducted a Pearson product-moment correlation analysis with the scores on study measures. To address H3, we conducted hierarchical regression analyses to examine whether dispositional mindfulness and self-compassion, entered in the last step of the model (step 3), would predict the variance in depression and anxiety symptoms over and above demographics (age and gender, step 1) and dispositional envy (step 2). Finally, to address H4, we conducted Fisher’s *r*-to-*z* transformations to examine the strength of correlations between envy and other measures, before and after controlling for mindfulness and self-compassion. All analyses were conducted on SPSS version 23 (IBM Corp., Armonk, NY, USA), and the alpha level of significance was set to 0.05.

## 3. Results

A total of 18.3% (*n* = 19) participants reported clinically significant depressive symptoms, and 23.1% (*n* = 24) reported clinically significant anxiety symptoms. The correlations among all measures are shown in Table 1.

Consistent with H1a, dispositional envy was significantly and negatively associated with job satisfaction (*r* = −0.20), and significantly and positively associated with stress (*r* = 0.67), depression (*r* = 0.57), and anxiety (*r* = 0.63). Accordingly, those who reported a higher capacity to experience envy tended to also report lower job satisfaction, higher symptoms of depression and anxiety, and higher perceived stress.

Consistent with H1b, we found that dispositional envy was associated with higher negative emotions (*r* = 0.67) and lower positive emotions (*r* = −0.23). Accordingly, those who reported a higher capacity to experience envy, also tended to report more intense negative and less intense positive emotions over the previous week. Consistent with H1c, dispositional envy was associated with lower subjective resilience (*r* = −0.47). Accordingly, those who reported a high capacity for envy tended to also report a perceived lower ability to “bounce back” after challenges.

Consistent with H2a, dispositional mindfulness was significantly and positively associated with job satisfaction (*r* = 0.33), and significantly and negatively associated with stress (*r* = −0.68), depression (*r* = −0.57), and anxiety (*r* = −0.53). Consistent with H2b, dispositional self-compassion was significantly and positively associated with job satisfaction (*r* = 0.36), and significantly and negatively associated with stress (*r* = −0.70), depression (*r* = −0.48), and anxiety (*r* = −0.53). Consistent with H2c, mindfulness and self-compassion were both associated with lower negative emotions, higher positive emotions, and subjective resilience. Accordingly, those who reported higher capacities for mindfulness and self-compassion, tended to also report improved indices of mental health and improved perceptions of resilience in the face of adversity.

Consistent with Chmitorz et al.’s [13] model, we expected that protective factors would predict resilience and associated outcomes over and above the risk factors. As set forth by H3, hierarchical linear regressions showed scores on the risk factor of envy entered in the second step of each model explained significant unique variance in job satisfaction (5%), stress (43%), depression (31%), and anxiety (38%), beyond variance attributed by age and gender. Scores on the protective factors of dispositional mindfulness and self-compassion entered in the last step, together uniquely and significantly predicted variance in job satisfaction (10%), stress (19%), depression (8%), and anxiety (4.5%), beyond variance attributed by demographics and dispositional envy (Table 2).

Consistent with H4, controlling for mindfulness and self-compassion significantly reduced correlations between dispositional envy and stress (*r* = 0.67 to 0.36; *z* = 3.05; *p* = 0.0023), depression (*r* = 0.57 to 0.34; *z* = 2.01; *p* = 0.044), and anxiety (*r* = 0.63 to 0.42; *z* = 2.07; *p* = 0.039). The correlation between envy and job satisfaction was also reduced, but not significantly (*r* = −0.20 to *r* = 0.062; *z* = −1.86; *p* = 0.063).

## 4. Discussion

PSP, including police trainees, experience several traumatic stressors in the line of duty. These experiences increase their risk of developing significant symptoms of mental disorder (e.g., depression and anxiety: [1]). Furthermore, police officers are trained in environments likely to trigger social comparisons, and such comparisons often lead to the painful emotion of envy. Envy, with its quintessential negative affectivity, opens the door for exacerbated mental health symptoms, as a consequence of encountering highly stressful situations. While few suggestive studies have found a link between increased envy and poorer mental health [5], even fewer have examined the interplay of this risk factor with transdiagnostic protective factors in predicting mental health symptoms. Accordingly, we examined whether dispositional envy was associated with job satisfaction, stress, and mental health in a sample of national police trainees. The results confirmed that envy is a pernicious risk factor, which is associated with significantly lower job satisfaction, increased stress, increased negative emotions, decreased positive emotions and perceptions of resiliency, and increased symptoms of depression and anxiety. Conversely, individual differences in two protective capacities—mindfulness and self-compassion—were associated with higher job satisfaction, lower stress, lower negative emotions, higher positive emotions and perceptions of resiliency, and fewer symptoms of depression and anxiety. The associations of individual difference variables (envy, mindfulness, and self-compassion) with health and occupational outcomes were large (characterized by effect sizes of *r* > 0.30: [41]). Hierarchical regression models further revealed that the protective factors of mindfulness and self-compassion, above and beyond envy, explained the significant variance in job satisfaction (10%), depression (8%), and anxiety (4.5%). Finally, the strength of associations between envy and the relevant outcomes significantly diminished after controlling for mindfulness and self-compassion, indicating a “buffering” effect of these protective traits.

A substantial minority of our sample reported clinically significant symptoms of depression and anxiety. This is consistent with the published literature that demonstrates increased symptoms of mental disorders among PSP, compared to members of the general population [1]. Our findings on the relationships of dispositional mindfulness and self-compassion with indices of mental health and job satisfaction are consistent with previously published trials, which demonstrate the protective role that such dispositional factors play in directly mitigating mental health outcomes and in bolstering coping capacities [19,20,21]. Furthermore, our findings demonstrated the link between envy and poorer mental health and a poorer overall emotional state. They also demonstrated how mindfulness and self-compassion may help to loosen this link and, hence, potentially confer resilience in times of adversity [14].

Our findings further highlight the potentially pernicious effects of dispositional malicious envy on mental health. We found that dispositional envy is not only directly associated with mental health symptoms, but was also associated with the experience of more intense negative emotions. This association points to the potentially dynamic nature of the relationship between envy, negative emotions, and poorer mental health; envy is associated with more intense negative emotions, which are, in turn, both a by-product and accelerant of mental health symptoms [42,43].

We found that while dispositional mindfulness and self-compassion scores were highly correlated, the correlation was low enough to suggest that they may be distinct but overlapping constructs [44]. Accordingly, these constructs could potentially represent unique and cultivatable skills. While mindfulness is a subcomponent of self-compassion, as conceptualized by Neff [17], this factor of self-compassion is focused narrowly on awareness and acceptance of one’s painful or negative experiences. The complete mindfulness construct is more broadly focused on awareness and acceptance of the full gamut of present-moment experiences, whether positive or negative [45]. Furthermore, mindfulness and self-compassion meditations appear to activate different brain regions, potentially signaling that they represent unique regulatory mechanisms, and, hence, could be targeted separately [46,47].

Finally, we found that envy was associated with lower positive emotions and subjective resilience. Conversely, dispositional mindfulness and self-compassion were associated with higher positive emotions and subjective resilience. Resilience, even when examined through self-report measures, has been found to correlate meaningfully with improved mental and physical health [11]. Increased positive emotions could be a key mechanism in understanding why mindfulness and self-compassion confer broad protection against ill health.

Although a nuanced discussion of the mechanisms of dispositional mindfulness and self-compassion is beyond the scope of this cross-sectional trial, the current findings are suggestive. Researchers have found that the relationship between subjective, relative deprivation—a process akin to envy, typified by social comparisons combined by feelings of inferiority and resentment toward the comparison target—and depression was fully mediated by negative, automatic thoughts about self [48]. Researchers have also found that both dispositional mindfulness and self-compassion were associated with lower negative automatic thoughts about self [49,50]. Accordingly, the buffering effects of dispositional mindfulness and self-compassion on mental health, found in the present trial, could be attributed to reduced negative automatic thoughts. Furthermore, both of the examined protective factors have been found to instigate a cascade of other mechanisms, such as lower rumination [51], and as partially demonstrated in this trial, increased positive emotions [52], and improved attentional regulation [53], which could explain both improved mental health and higher job satisfaction. Future research with different PSP should confirm the mediation status of negative self-referent thoughts and positive emotions in explaining the downstream effects of trait mindfulness and self-compassion.

To the author’s knowledge, the current study is the first to examine the transactional nature of risk factors, such as envy, with protective factors, such as dispositional mindfulness and self-compassion, among PSP. Accordingly, and given the relatively few studies on the role of envy in mental health [10], the results of the current study fill a relatively important knowledge gap in this high-risk population.

This study had limitations that can provide directions for future research. First, the sample size of the study was relatively small; there were inherent difficulties in recruiting participants from such a hard-to-access population, who have little discretionary time, so caution is necessary when interpreting these results. Second, although the present sample appeared to be representative of the overall population of the RCMP officers [1], there also appeared to be some key differences. For example, the current study sample was comprised, to a large extent, of officers who identify as men, and who have European ancestry. Consequently, results may not generalize to all first responders (or to all employees more generally). Future research should examine the reliability of the obtained findings among a more culturally diverse sample of PSP. Third, the cross-sectional design of this study limited any causal inferences (though such designs are more useful than often claimed: [54]). For example, the present study demonstrated relationships between individual difference constructs, such as self-compassion, mindfulness, and envy, with symptoms of mental health and job satisfaction; however, the directionality of influence among such factors and the potential influence of additional, presently unexamined variables, are not clear.

## 5. Conclusions

The results of this study further demonstrate the protective natures of dispositional mindfulness and self-compassion, which work beyond the diagnostic categories of depression and anxiety. These capacities appeared to attenuate the pernicious effects of dispositional envy among a sample of police trainees in Canada, with envy being a woefully neglected emotion in mental health research generally, and among this population specifically. Furthermore, this study has highlighted the importance of examining risk factors together with protective factors, especially in domains involving emotions and their regulation (or dysregulation) [13]. This dynamic conception has highlighted that resilience can be fostered via interventions targeted at the reduction of the effects of risk factors (e.g., reduced envy), or at the cultivation of protective factors (e.g., mindfulness and self-compassion). Given the malleability of these protective constructs, even with low intensity and cost-effective interventions [55], this approach may provide efficient solutions to enhancing job-related attitudes and PSP’s mental health. Even brief, self-guided mindfulness-based interventions, which are highly scalable and easy to administer, appear to be efficacious in improving several mental health indices [56]. These interventions encourage their trainees to participate regularly in simple exercises (e.g., writing a self-compassionate letter, and giving yourself a compassionate break: [57], and meditations (body scan and loving-kindness meditation). Even these simple interventions, that require low or minimal therapist guidance, have been shown to cultivate the skills of mindfulness and self-compassion, which, in turn, appear to protect individuals across diagnostic categories [58]. Such low-intensity interventions lend themselves well to dynamic and uncertain work environments, such as those encountered by PSP.

## Figures and Tables

**Table 1 ijerph-19-05926-t001:** Pearson correlations between job attitudes, risk factors, and protective factors.

	M (SD)	α	JobSat	PHQ-8	GAD-7	PNAS-N	PNAS-P	BRS	FFMQ	SCS	DES
**JobSat**	15.5 (3.5)	0.81	—								
**PHQ-8**	5.5 (4.5)	0.85	−0.22 *	—							
**GAD-7**	6.1 (5.1)	0.92	−0.22 *	0.77 **	—						
**PNAS-N**	11.0 (3.7)	0.75	−0.25 **	0.66 **	0.73 **	—					
**PNAS-P**	19.1 (3.3)	0.79	0.38 **	−0.43 **	−0.24 *	−0.38 **	—				
**BRS**	25.9 (5.8)	0.78	0.29 **	−0.35 **	−0.42 **	−0.50 **	0.45 **	—			
**FFMQ**	52.4 (7.9)	0.77	0.33 **	−0.57 **	−0.53 **	−0.63 **	0.62 **	0.53 **	—		
**SCS**	39.7 (8.1)	0.87	0.36 **	−0.48 **	−0.53 **	−0.66 **	0.54 **	0.68 **	0.71 **	—	
**DES**	18.4 (10.1)	0.92	−0.20 *	0.57 **	0.63 **	0.67 **	−0.23 *	−0.47 **	−0.57 **	−0.63 **	—
**PSS**	15.3 (7.0)	0.86	−0.27 **	0.69 **	0.73 **	0.81 **	−0.51 **	−0.57 **	−0.68 **	−0.70 **	0.67 **

Notes: * *p* < 0.05; ** *p* < 0.01. M—mean; SD—standard deviation; JobSat—Job Satisfaction, as measured by the Job Descriptive Index; PHQ-8—Patient Health Questionnaire (depression); GAD-7—Generalized Anxiety Disorder scale; PNAS-N—Positive and Negative Affect Schedule Short Form–Negative; PNAS-P—Positive and Negative Affect Schedule Short Form–Positive; BRS—Brief Resilience Scale; FFMQ—Five-Factor Mindfulness Questionnaire; SCS—Self-Compassion Scale Short Form; DES—Dispositional Envy Scale; PSS—Perceived Stress Scale.

**Table 2 ijerph-19-05926-t002:** Hierarchical regression model of demographics (age and gender), risk factor (dispositional envy), and protective factors (mindfulness and self-compassion) on (a) job satisfaction; (b) stress; (c) depression; and (d) anxiety.

Job Satisfaction	*B*	*SE*	*β*	*t*	*p*
Step 1 (*Demographics*): *R* = 0.17, *R*^2^ = 0.029					
Age	−0.10	0.05	−0.20	−2.12	0.04
Gender	0.23	0.75	0.03	0.31	0.76
Step 2 (*Risk Factor*): *R* = 0.28, *R*^2^ = 0.078, Δ*R*^2^ = 0.05 *					
DES	0.01	0.04	0.04	0.35	0.73
Step 3 (*Protective Factors*): *R* = 0.42, *R*^2^ = 0.18, Δ*R*^2^ = 0.10 **					
FFMQ	0.08	0.06	0.17	1.27	0.21
SCS	0.12	0.06	0.28	1.90	0.06
**Stress**					
Step 1 (*Demographics*): *R* = 0.20, *R*^2^ = 0.039					
Age	0.002	0.06	0.002	0.03	0.98
Gender	1.84	1.01	0.11	1.82	0.07
Step 2 (*Risk Factor*): *R* = 0.32, *R*^2^ = 0.47, Δ*R*^2^ = 0.43 ***					
DES	0.21	0.06	0.30	3.76	<.001
Step 3 (*Protective Factors*): *R* = 0.46, *R*^2^ = 0.66, Δ*R*^2^ = 0.19 ***					
FFMQ	−0.28	0.08	−0.31	−3.52	0.001
SCS	−0.27	0.08	−0.30	−3.26	0.002
**Depression**					
Step 1 (*Demographics*): *R* = 0.15, *R*^2^ = 0.021					
Age	−0.02	0.05	−0.03	−0.34	0.73
Gender	−0.09	0.84	−0.01	−0.11	0.91
Step 2 (*Risk Factor*): *R* = 0.57, *R*^2^ = 0.33, Δ*R*^2^ = 0.31 ***					
DES	0.16	0.05	0.37	3.54	0.001
Step 3 (*Protective Factors*): *R* = 0.64, *R*^2^ = 0.41, Δ*R*^2^ = 0.081 **					
FFMQ	−0.20	0.07	−0.35	−3.04	0.003
SCS	0.004	0.07	0.01	0.06	0.96
**Anxiety**					
Step 1 (*Demographics*): *R* = 0.19, *R*^2^ = 0.037					
Age	−0.07	0.06	−0.09	−1.15	0.26
Gender	−0.61	0.92	−0.05	−0.66	0.51
Step 2 (*Risk Factor*): *R* = 0.64, *R*^2^ = 0.42, Δ*R*^2^ = 0.38 ***					
DES	0.23	0.05	0.45	4.45	<0.001
Step 3 (*Protective Factors*): *R* = 0.68, *R*^2^ = 0.46, Δ*R*^2^ = 0.045 *					
FFMQ	−0.11	0.07	−0.16	−1.47	0.15
SCS	−0.09	0.07	−0.14	−1.19	0.24

Notes: * *p* < 0.05; ** *p* < 0.01; *** *p* < 0.001; asterisks indicate statistical significance for model change.

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
