# Peer review of "Resilience in the Ranks: Trait Mindfulness and Self-Compassion Buffer the Deleterious Effects of Envy on Mental Health Symptoms among Public Safety Personnel"

_ijerph, 2022, doi:10.3390/ijerph19105926_

Round 1
Reviewer 1 Report
- Are you using APA 6 to 7? If 7 all authors in articles with 3 or more begin as author, et al- even the first time article is used.
- If APA 6- Line 72 you have a citation of 6 authors. Per APA 6- 6 or more are et al from first article.
- Line 100- you state there is little on Self compassion in the workplace. If you are citing criminal justice perhaps this is true, but if all workplace- see these and other articles. Much research has been done on SC and work
Hashem, Z & Zeinoun, P. (2020). Self compassion explains less burnout among healthcare professions. Mindfulness.
Miller, Lee, Shalash, Poklembova (2020)…Self compassion among social workers Journal of Social Work
Devenesh-Mears, P. Call to compassionate self care… Journal of Spirituality and mental health (2015) Vol 17 issue 1
- Line 113- Transdiagnostic. Spelling error of Transdiacnostic
- Line 198-200. Given the results that many participants rated high for Depression and Anxiety, did the researchers have a plan to assist participants with these concerns? Were participants provided assistance contact information at the time the survey was conducted. A chance was taken with this research that you would come to know that trainees were experiencing mental health issues so what response measures were offered for help? If no such measures were made or considered prior to launching this study, please state why.
- In conclusions you write about this study lending to the knowledge that “This dynamic conception highlights that resilience can be fostered via interventions targeted at reduction of the effects of risk factors (e.g., reduced envy), or cultivation of protective factors (e.g., mindfulness and/or self-compassion). Given the malleability of these protective constructs, even with low intensity and cost-effective interventions (Beshai et al., 2020), this approach may provide efficient solutions to enhancing job-related attitudes and PSP mental health.” Please provide some of your ideas for examples of such interventions. it will make this section even more impactful.
- This article is appropriate for publication consideration if the above feedback is well attended to. This study is needed for the PSP realm, but also needs more focus on proposed interventions to increase trainee self compassion and mindfulness.
Author Response
Reviewer 1 Comments |
Authors’ Response |
Page Reference |
|
We thank the reviewer for this correction. We have now fixed all in-text citations with 3+ authors to reflect APA 7th edition guidance (e.g., use of “et al.” when citing articles with 3+ names, even on first use). |
P. 2-4 |
2. Line 100- you state there is little on Self compassion in the workplace. If you are citing criminal justice perhaps this is true, but if all workplace- see these and other articles. Much research has been done on SC and work Hashem, Z & Zeinoun, P. (2020). Self compassion explains less burnout among healthcare professions. Mindfulness. Miller, Lee, Shalash, Poklembova (2020)…Self compassion among social workers Journal of Social Work Devenesh-Mears, P. Call to compassionate self care… Journal of Spirituality and mental health (2015) Vol 17 issue 1 |
We agree with the reviewer that self-compassion has been examined more than we proclaimed in the previous iteration of the manuscript. Accordingly, we have now corrected this statement in Line 100, and now include references to the important works suggested by the reviewer (Devenesh-Mears et al., 2015; Hashem & Zeinoun, 2020; Miller et al., 2020). The corrected line now reads: “Self-compassion has been studied less frequently than mindfulness in the workplace; however, the available evidence suggests self-compassion is associated with improved satisfaction and functioning in diverse workplace settings (Devenesh-Mears, 2015; Hashem & Zeinoun, 2020; Miller et al., 2020). Self-compassion has been studied even less frequently among police and other PSP, a gap we shed light on here.” |
P. 3 Lines 118-123 |
|
Thank you for catching this typo. We have corrected the spelling error in the revised manuscript |
P. 3, Line 144 |
|
All participants in the study were provided psychoeducation regarding depression, stress, mindfulness, and self-compassion. All participants were also offered the opportunity to partake in guided mindfulness and self-compassion meditations and exercises. Finally, all participants were provided a list of evidence-based self-help resources for anxiety and depression, as well as a list of local mental health resources they can tap into at their own volition. We now provide this information under the procedure subsection of the revised manuscript: “After completing questionnaires, participants were offered the opportunity to watch a psychoeducational video on depression, anxiety, and stress. Participants were also offered links to online guided mindfulness meditations, as well as self-compassion exercises and techniques. Finally, at the end of the trial all participants were offered a list of evidence-based self-help resources as well as contact information for local mental health services.“
|
P. 4 Lines 199-204. |
|
We thank the reviewer for this comments. We have now added a brief section in the conclusion subsection discussing examples of simple exercises and meditations that have been demonstrated to cultivate mindfulness and self-compassion: “Even brief, self-guided mindfulness-based interventions, which are highly scalable and easy to administer, appear to be efficacious in improving several mental health indices (Taylor et al., 2021). These interventions encourage their trainees to participate regularly in simple exercises (e.g., writing a self-compassionate letter; giving yourself a compassionate break; Germer & Neff, 2019) and meditations (body scan; loving-kindness meditation). Even these simple interventions, requiring low or minimal therapist guidance have been demonstrated to cultivate skills of mindfulness and self-compassion, which in turn appear to protect across diagnostic categories (Beshai et al., 2016).”
|
P. 10 Lines 461-470 |
|
We thank the reviewer for this helpful review and encouragement. |
NA |

Reviewer 2 Report
The work addresses an interesting and current topic such as the study of protective stress factors.
I believe that the research has been carried out correctly and its limitations have been indicated at the end of the text, with which I agree. Likewise, the results, discussion and conclusions are also consistent with the approaches and hypotheses.
In my opinion, to improve the publication of this research could be completed with the following aspects:
- The title should reflect more clearly the sample used in the research.
- It should specify whether the sample is representative of the group investigated.
- It does not clarify the type of sampling used to establish the sample. Nor how the participants in the research were recruited.
- The procedure followed in the study is not clearly described.
- The date on which the investigation was conducted or the data was collected is not indicated.
- The discussion talks about variables that were not the objective of the research.
Author Response
Reviewer 2 Comments |
Authors’ Response |
Page Reference |
The work addresses an interesting and current topic such as the study of protective stress factors. I believe that the research has been carried out correctly and its limitations have been indicated at the end of the text, with which I agree. Likewise, the results, discussion and conclusions are also consistent with the approaches and hypotheses. |
We thank the reviewer for these encouraging words and quality review. We agree that the paper addresses an important and neglected topic, and we believe that the results contribute meaningfully to the literature. |
NA |
In my opinion, to improve the publication of this research could be completed with the following aspects: 1. The title should reflect more clearly the sample used in the research. |
We have now revised the title to more clearly reflect the sample and methodology employed. The title now reads “Resilience in the ranks: trait mindfulness and self-compassion buffer the deleterious effects of envy on mental health symptoms among Public Safety Personnel” |
P. 1 |
2. It should specify whether the sample is representative of the group investigated.
|
We now briefly qualify the representativeness of the sample in the limitations subsection of the discussion section: “Second, and although the present sample appears representative of the overall population of Royal Canadian Mounted Police officers (Carleton et al., 2018), there appears to be some key differences. For example, the current study sample was comprised to large extent of officers identifying as men with European ancestry.” |
P. 9 Lines 428-431 |
3. It does not clarify the type of sampling used to establish the sample. Nor how the participants in the research were recruited.
|
We thank the reviewer for this comment. We have added much more detail related to participant recruitment, design, and sample characteristics. For example, the revise manuscript now includes “In the present study, we employed convenience sampling and a cross-sectional design.” |
P. 4, Line 184 |
4. The procedure followed in the study is not clearly described.
|
The revised manuscript now includes the following to clarify study procedure: “Participants were in their first or second week of a multi-week training regimen when they completed the measures. Participants were administered the questionaries in groups of 25-32 trainees, in Internet-enabled computer rooms on the RCMP training campus.” |
P. 4, Lines 186-189 |
5. The date on which the investigation was conducted or the data was collected is not indicated. |
We now provide this information:
|
P. 4, Line 204 |
6. The discussion talks about variables that were not the objective of the research. |
We thank the reviewer for this comment. We have substantially revised the discussion section to only discuss variables that were most pertinent to the study. |
P. 8-10. |

Reviewer 3 Report
Method
1. Participants - The authors only provided the sample size, sex of participants, and age of the sample. Is there more information to further contextualize and describe the sample? Gender identity, sexual orientation, social class, and etc. are a few demographics that would be helpful to understand the sample.
2. The data was collected during Fall 2020. How did the authors account for depression and anxiety related to the COVID-19 pandemic?
Discussion
- The authors should also comment on how the use of a cross-sectional design precludes inferring causation. Also, could there be there could be bidirectionality among the study variables? What about respondent bias?
- This section could benefit a lot from a more thorough and in-depth discussion of some important topics regarding how five-factor mindfulness and self-compassion works. For example, how do the authors think about the roles of the different components of mindfulness and self-compassion?
In some cases, self-compassion scale includes mindfulness. For example, the Self-Compassion Scale (Neff, 2003) comprises 26 items with six subscales to measure three general facets of self-compassion: self-kindness versus self-judgment, common humanity versus isolation, and mindfulness versus over-identification.
Please provide rationale/previous studies to support using mindfulness and self-compassion together in this study.
- The authors could add any acknowledgment of cultural factors or experiences that may explain study findings.
Author Response
Reviewer 3 Comments |
Authors’ Response |
Page Reference |
|
We thank the reviewer for these very helpful suggestions and comments. We have added more information describing and contextualizing the sample as suggested. For example, the revised manuscript now reads: “Of the 173 participants receiving the survey, 104 (25 identifying as women, 79 identifying as men; Mage = 29.27, SDage = 6.92) provided their informed consent and responses to the majority of scale items and were retained for analyses. Participants were excluded if they failed to respond to 20% or more of study items. A total of 46 participants (44.2%) identified as European, 3 as Korean, 2 as Persian, and 2 as Punjab, with various other identifications (e.g., Russian; Japanese, Latino). The majority (n = 57; 55%) of participants reported having tried meditation at least once. Of those who tried meditation, 70% reported their meditation skill level as novice, 26.5% as intermediate, and 3.5% as advanced.” |
P. 4, 191-198 |
2. The data was collected during Fall 2020. How did the authors account for depression and anxiety related to the COVID-19 pandemic? |
While the ethics certification renewal date is Fall 2020, data collection for this study started in Spring 2018 and ended summer 2018. We have added this information to the appropriate section. Accordingly, this was several years prior to the COVID-19 pandemic. |
P. 4, Lines 204 |
Discussion
|
We have now included additional limitations in the limitations subsection of the discussion section. We have accordingly added the suggested information: “Third, the cross-sectional design of this study limits any causal inferences (though such designs are more useful than often claimed; Spector, 2019). That is, the present study demonstrated relationships between individual-difference constructs such as self-compassion, mindfulness, and envy with symptoms of mental health and job satisfaction; however, the directionality of influence among such factors and the potential influence of additional, presently unexamined variables, are not clear.”
|
P. 9, 434-439 |
2. This section could benefit a lot from a more thorough and in-depth discussion of some important topics regarding how five-factor mindfulness and self-compassion works. For example, how do the authors think about the roles of the different components of mindfulness and self-compassion? In some cases, self-compassion scale includes mindfulness. For example, the Self-Compassion Scale (Neff, 2003) comprises 26 items with six subscales to measure three general facets of self-compassion: self-kindness versus self-judgment, common humanity versus isolation, and mindfulness versus over-identification. Please provide rationale/previous studies to support using mindfulness and self-compassion together in this study.
|
We thank the reviewer for these important and valuable suggestions. We were very persuaded by the reviewer’s comments to the extent of adding three additional, previously unpublished variables from the present trial into the current manuscript. The additional variables are positive emotions (PNAS-P); negative emotions (PNAS-N); and subjective resilience (BRS). We believe adding these three additional variables into the correlation matrix provided ample opportunity to discuss potential mechanism and the “why” and “how” of our study: “Our findings further highlight the potentially pernicious effects of dispositional malicious envy on mental health. We found dispositional envy is not only directly associated with mental health symptoms, but also associated with the experience of more intense negative emotions. This association points to the potentially dynamic nature of the relationship between envy, negative emotions, and poorer mental health; envy is associated with more intense negative emotions, which are in turn both a by-product and accelerant of mental health symptoms (Gallo & Matthews, 1999; Young et al., 2019). We found that while dispositional mindfulness and self-compassion scores were highly correlated, the correlation was low enough to suggest they are distinct but overlapping constructs (Henseler et al., 2015). Accordingly, these constructs could potentially be representing unique and cultivatable skills. While mindfulness is a subcomponent of self-compassion as conceptualized by Neff (2003), this factor of self-compassion is focused narrowly on awareness and acceptance of one’s painful or negative experiences. The complete mindfulness construct is more broadly focused on awareness and acceptance of the full gamut of present-moment experiences, whether positive or negative (Beshai et al., 2018). Further, mindfulness and self-compassion meditations appear to activate different brain regions, potentially signalling that they represent unique regulatory mechanisms, and hence could be targeted separately (Fletcher et al., 2010; Guan et al., 2021).
|
P. 8, Lines 375-392 |
|
We now discuss the lack of cultural diversity as a potential limitation in the limitations subsection of the discussion section. “Second, and although the present sample appears representative of the overall population of RCMP officers (Carleton et al., 2018), there appears to be some key differences. For example, the current study sample was comprised to large extent of officers identifying as men with European ancestry. Consequently, results may not generalize to all first responders (or employees more generally). Future research should examine the reliability of the obtained findings in a more culturally diverse sample of PSP.” |
P. 9, Lines 426-431. |

Reviewer 4 Report
Dear authors, thank you for the opportunity to get acquainted with your work.
To improve the presentation of your research results, you need to make some changes:
1. Expand the description of methods and techniques, indicate the author, provide links to publications of authors-developers.
2. The results start by presenting the mean values ​​and standard errors of the mean for the key parameters that are analyzed in the articles. And also to provide a description of the qualities level and their interpretation in accordance with the specifics of professional activity.
3. Add to the results not only a listing of the obtained connections, but also a description how these interrelated features can manifest themselves in the activities of employees (what they can interfere with or help).
4. In the results discussion, add how the study results can be applied in practice. What authors see directions for practical work with these employees. Also, to supplement the correlation of the obtained results with similar ones carried out on similar samples.
These comments do not detract from the overall positive impression of the work.
Best regards, Reviewer
Author Response
Reviewer 4 Comments |
Authors’ Response |
Page Reference |
1. Expand the description of methods and techniques, indicate the author, provide links to publications of authors-developers. |
We thank the reviewer for these very thoughtful suggestions and comments. We have added more information describing and contextualizing the sample as suggested. For example, the revised manuscript now reads: “In the present study, we employed convenience sampling and cross-sectional design. Survey links were provided to 173 trainees in the Royal Canadian Mounted Police (RCMP), which is the federal police service of Canada. Trainees completed measures online in randomized order as part of a larger mindfulness trial. Participants were in their first or second week of a multi-week training regimen when they completed the measures. Participants were administered the questionaries in groups of 25–32 trainees, in Internet-enabled computer rooms on the RCMP training campus.” |
P. 4, 183-189 |
1. The results start by presenting the mean values ​​and standard errors of the mean for the key parameters that are analyzed in the articles. And also to provide a description of the qualities level and their interpretation in accordance with the specifics of professional activity. |
We thank the reviewer for these valuable suggestions In Table 1, we now present means and standard deviations of key study variables. We also now provide a guide as to how to interpret scores on the key measures: “Scale scores were tabulated so that higher scores on each of the measures are indicative of greater levels of the construct assessed.“ |
P. 6., Lines 275; P. 5 Lines 258-259. |
2. Add to the results not only a listing of the obtained connections, but also a description how these interrelated features can manifest themselves in the activities of employees (what they can interfere with or help). |
Consistent with this suggestion, we now provide some descriptions of how to interpret findings in the results section: “Accordingly, those reporting a higher capacity to experience envy tended to also report lower job satisfaction, higher symptoms of depressing and anxiety, and higher perceived stress.” |
P. 6, Lines 287-289 |
3. In the results discussion, add how the study results can be applied in practice. What authors see directions for practical work with these employees. Also, to supplement the correlation of the obtained results with similar ones carried out on similar samples.
|
We again thank the reviewer for these very helpful suggestions. We have now offered more practical directions in accordance with trial findings which can be found in the discussion section: “These interventions encourage their trainees to participate regularly in simple exercises (e.g., writing a self-compassionate letter; giving yourself a compassionate break; Germer & Neff, 2019) and meditations (body scan; loving-kindness meditation). Even these simple interventions, requiring low or minimal therapist guidance have been demonstrated to cultivate skills of mindfulness and self-compassion, which in turn appear to protect across diagnostic categories (Beshai et al., 2016). Such low intensity interventions lend themselves well to dynamic and uncertain work environments such as those encountered by PSP. “
|
P. 9-19, Lines 460-470. |
